# Parametric Formula for Stress Concentration Factor of Fillet Weld Joints with Spline Bead Profile

**DOI:** 10.3390/ma13204639

**Published:** 2020-10-17

**Authors:** Yixun Wang, Yuxiao Luo, Seiichiro Tsutsumi

**Affiliations:** 1Joining and Welding Research Institute, Osaka University, Osaka 567-0047, Japan; wang.yixun@jwri.osaka-u.ac.jp; 2Department of Structural Engineering, Tongji University, Shanghai 200092, China; 1410210@tongji.edu.cn

**Keywords:** parametric formulae, stress concentration factor, fillet weld, weld model, probability analysis

## Abstract

The existing parametric formulae to calculate the notch stress concentration factor of fillet welds often result in reduced accuracy due to an oversimplification of the real weld geometry. The present work proposes a parametric formula for the evaluation of the notch SCF based on the spline weld model that offers a better approximation of the real shape of the fillet weld. The spline model was adopted in FE analyses on T-shape joints and cruciform joints models, under different loading conditions, to propose a parametric formula for the calculation of the SCF by regression analysis. In addition, the precision of parametric formulae based on the line model was examined. The magnitude of the stress concentration was also analyzed by means of its probability distribution. The results show that the line model is not accurate enough to calculate the SCF of fillet weld if the weld profile is considered. The error of the SCF by the proposed parametric formulae is proven to be smaller than 5% according to the testing data system. The stress concentration of cruciform joints under tensile stress represents the worst case scenario if assessed by the confidence interval of 95% survival probability.

## 1. Introduction

The welded joints have been widely used in thin-walled structures such as bridges, oil rigs, pressure vessels and ships, due to their low cost, superior performance, and high reliability [1,2], Fatigue cracks are prone to occur at the welded joint under cyclic loadings, which significantly threaten the safe operation of the thin-walled structures [3,4]. The fatigue behavior of welded joints is susceptible to several factors, including the residual stress, loading magnitude, loading type, and local stress concentration [5,6]. Fatigue cracks usually initiate in correspondence of stress localizations caused by geometric discontinuities and notches [7]. Therefore, it is important to assess the stress concentration of welded joints for both accurate prediction of their fatigue life and to propose a geometrical shape of the weld (for example, by grinding [8]) that could improve the fatigue performance.

The stress concentration effects can be expressed by the stress concentration factor (SCF), which has been widely applied in the engineering structures [9]. Several methods are available to calculate the SCF including analytical, numerical, and experimental analyses. The analytical approach, based on elastic mechanics, is usually applied to some simple geometric shapes [10], while it is difficult to use to assess the SCF of weld with sophisticated shapes of weld toe or root. The photo-elastic method [11] is one of the experimental ways to assess the SCF by visualizing the elastic stress directly on the welded components. On the other hand, the accuracy of the estimation of the SCF with the photo-elastic method remains to be improved. In recent years, some new technologies, such as the fiber Bragg gratings [12], have been applied to assess the stress concentration in fatigue experiments with promising results. A wider diffusion of these new technologies is expected in the future. Numerical methods, based on finite elements (FE) [13], have usually been used to obtain parametric formulae of the SCF. This approach allows to calculate a SCF as a function of essential geometric parameters of the weld (weld toe radius *r*, flank angle *α*, etc.). Then parametric formulae can be fitted by selecting suitable numerical functions [14]. The neural network method [15], developed in recent years, is also available to ensure a high accuracy in the estimation of the SCF. A variety of formulae considering key geometric parameters have been proposed based on the numerical methods considering different types of welded joints, including the cruciform joints [16], T-shape joints [17], lap joints [18], and butt joints [19]. Modifications of the aforementioned formulae have also been carried out in order to address particular cases such as loading types (bending load and tensile load) [20], weld shape after grinding [21], additional weld [22], and penetration ratio [23].

An accurate weld model is necessary in the FE analyses for the precise computation of SCF. The existing parametric formulae are mostly based on the line model, i.e., the shape of the weld surface is usually approximated with a straight line connecting the arc of the weld toe and the upper weld end [24]. The flank angle and weld leg length were considered while the streamline shape of the weld was neglected. The predicted results can be acceptable under some conditions if the weld shape resembles a straight line. However, the weld shape is usually arc-shaped and its approximation with a straight line can result in reduced accuracy of the SCF. The real shape of the weld can be obtained by 3D laser scanning technology. For instance, Hou [25] applied this technology to acquire the real weld toe geometry of cruciform specimens and to calculate the SCF based on FE analyses. However, the relationship between the geometric parameters and the SCF has not been analyzed yet, and the parametric formula for general conditions is not proposed.

The novelty of the work consists in the formulation of a parametric formula for the fillet weld of the T-shape joint and cruciform joint based on the FEM and regression analysis. The geometric model with a spline curve of the weld is proposed to consider the real shape of the weld. The accuracy of the line model widely used in the existing parametric formulae is also discussed. Subsequently, based on a large set of FE analysis, the existing parametric formulae for the fillet weld are tested and discussed. The influence of key parameters on the SCF is investigated, and the parametric formulae for the T-shape joint and cruciform joint under the tensile stress and bending stress are proposed through the training data system and are examined by the testing data system. The parameters of a real specimen were also measured, and the severity of stress concentration is assessed by means of a probabilistic approach.

## 2. Finite Element Analysis

### 2.1. Proposed Spline Model

The line model has been widely used to evaluate the SCF at the weld toe of the fillet weld joints. A typical example of a fillet weld model (T-shape joint), adopting the line method, is reported in Figure 1a. The main geometric features can be summarized as: the thickness of plate (*t*) and stiffener (*T*), the flank angle (*α*), the thickness of weld throat (*L*), and radius (*r*) of weld toe. The parameters mentioned above usually represent the variables that have the most significant influence on the SCF [26]. A figure taken for a real fillet weld is also provided for comparison [27]. It could be observed that the line model neglects to consider the profile of the welding and a straight line is used to approximate the real shape. The SCF, calculated based on the line model, could lack precision, especially when the hump of the weld is relevant. Thus, it is necessary to simulate the shape of the weld as accurately as possible for better precision.

A spline model which considered the shape of butt weld joints was suggested by Luo [19], while the proposed parametric formula could not be used to calculate the SCF of fillet weld. Hence, the spline model considering the profile of fillet weld is proposed as schematically reported in Figure 1b. The geometric shape of the spline model is based on the line model. Additional parameters are proposed including the radius (*r*_2_) and flank angle (*θ*_2_) of the weld toe on the topside, salient point position (*1/n*) and the hump height (*H*). The influence of top weld toe radius (*r*_2_) was considered for the integrity of the weld model. The top flank angle (*θ*_2_) defines the stretch direction of the spline from the stiffener. The salient point (*1/n*) and hump height (*H*) indicates the apex position of the spline compared to the line model. The spline starts at the end of the bottom arc, passes through the salient point and stops at the end of the top arc. Clearly, the proposed spline model exhibits a better approximation of the real shape of the fillet weld compared to the line model. The FE analyses presented in this work have been carried out with the spline approach.

### 2.2. Finite Element Model for Welded Joints

In the following sections, the parametric formulae for the SCF of T-shape welded joints and cruciform welded joints under the tensile and bending stresses are investigated. Here, the FE models for the joints are briefly discussed. A one-quarter model and a one-half model, with different boundary conditions, were employed to calculate the SCF at the weld toe of fillet weld considering different structures and loading conditions (see Figure 2). A unitary tensile stress is applied at the extremity of the plate acting as nominal (membrane) stress (*σ_t_*) in case of tensile loading conditions. Similarly, a unitary normal stress, linearly distributed on the section, is applied as nominal (bending) stress (*σ_b_*) in the case of bending load conditions.

The FE models for fillet weld were created by means of the commercial software ABAQUS v6.14-3. The material was assumed to be isotropic with an elastic modulus E = 206 GPa and a Poisson’s ratio *υ* = 0.3. Eight-node plane strain element CPE8 were used to increase the precision of the results. An example of global model and local meshes are shown in Figure 3a–b. Mesh sensitivity analyses were carried out considering different mesh discretizations in correspondence of the weld toe. Figure 3c reports an example of the mesh sensitivity analyses carried out for a cruciform joint under tensile stress with *t* = 10 mm, *T* = 10 mm, *r*_1_ = 0.43 mm, *r*_2_ = 0.43 mm, *θ*_1_ = 60°, *θ*_2_ = 60°, *L*_1_ = 8 mm, *L*_2_ = 8 mm, *n* = 3, and *H* = 1. The figures show the contour fields of the maximum principal stress (*σ_notch_*) at the weld toe. A mesh discretization with 10 more elements on the weld toe is sufficient to obtain errors smaller than 0.17%. However, it should be noted that the error of SCF depends on the number of elements on the weld toe, and the element number which is sufficient for favorable precision on the weld toe with small arc length might result in poor accuracy on that with greater arc length. Therefore, the convergence analysis is conducted based on the range of arc length studied in this paper (the range is discussed in Section 2.4). The *σ_notch_* at the weld toe for the maximum, minimum, and middle arc length, is calculated and normalized by the reference stresses *σ_notch,ref_* calculated by finite element model with very fine mesh (50 elements on the weld toe). It could be observed in Figure 3d that the calculated notch stress was converged when the element number on the weld toe is greater than 10. Even the error committed using an element number of 4 is still quite negligible and around 0.6%. In the following analyses, a mesh discretization with a minimum element number of 10 on the weld toe is used to ensure accurate results.

### 2.3. Comparison between the SCF Calculated by the Spline Model and by the Line Model

The results of the SCF at the weld toe obtained by the spline and the line approaches were compared to discuss the necessity to consider a better approximation of the weld shape. Taking the cruciform joint under tensile stress as an example, the essential parameters to build the fillet weld of the specimen were measured by the 3D Scanner VL-300, as shown in Figure 4a. Ten sections of the fillet weld were randomly cut from the whole specimen. Subsequently, FE models based on the spline and line models were created. The maximum principal stress at the weld toe surface was considered as the notch stress (*σ_notch_*). The elastic notch stress concentration factor at the weld toe of fillet weld was defined as in Equation (1)
(1)$$K_{t}=\sigma_{notch}/\sigma_{nom}$$
(2)$$\delta_{1}=\left(K_{t,spline}-K_{t,line}\right)/K_{t,spline}$$

Since the spline model offers a better approximation of the real shape of the fillet weld compared to the line model, the SCF calculated by the spline model was assumed as reference. Taking ±5% deviation (*δ*_1_ (%), Equation (2)) as the limit error, the results of the SCF for the spline model and line model are shown in Figure 4b. Only 30% of the SCFs calculated by the line model are included within the ±5% limit error (i.e., blue-dashed lines in Figure 4b). Moreover, the SCF calculated by the line model is always smaller than the one obtained by the spline model. Therefore, it is concluded that a judgment based on the line model could lead to an underestimation of the stress concentration.

### 2.4. Application Ranges

The precision of the SCF parametric formulae should be evaluated within the corresponding application ranges. According to former researches [19], the SCF at the weld toe is independent of the absolute values of the parameters; however, it is influenced by their relative ratios. Hence, the normalized parameter ranges were determined for the subsequent parametric formula as listed below:(1)Stiffener thickness *T/t*: 0.3–2.0;(2)Bottom weld toe radius *r*_1_/*t*: 0.003–0.36;(3)Top weld toe radius *r*_2_/*t*: 0.003–0.36;(4)Bottom flank angle *θ*_1_: 20°–90°(5)Top flank angle *θ*_2_: 20°–90°(6)Bottom weld leg length *L*_1_/*t*: 0.3–2.0;(7)Top weld leg length *L*_2_/*t*: 0.3–2.0;(8)Salient point position *1/n*: 0.2–0.9; and(9)Hump height *H/t*: 0.0–0.3.

The above parameter ranges were extended compared to the former parametric formulae to cover as many fillet weld shapes as possible. The ranges of the additional parameters for the spline model (i.e., top weld toe radius *r*_2_, top flank angle *θ*_2_, salient point position *1/n* and hump height *H*/t) were defined based on the measures obtained from real specimens.

## 3. Proposed Parametric Formula for Fillet Weld

### 3.1. Overview of Existing SCF Formulae

Most of the previous studies were conducted based on the line model and a lot of parametric formulae were proposed based on different loading conditions and welded joints [16,17,20,24,26,28,29,30,31,32]. Table 1 lists a few examples of the works where the four conditions in Figure 2a–d were investigated. The relevant formulae are summarized in Appendix A.

The weld models, based on the formulae reported in Table 1, are shown in the following Figure 5a–d. All these parametric formulae are based on the line model approach. It is worth mentioning that some of the weld parameters in these formulae have a slightly different definition from the ones in this paper. For example, the influence of *L*_1_/*t* and *T/t* on the SCF is considered by the parameter T_0_/t defined by Brennan et al. [17] as seen in Figure 5a. Besides, the influence of some parameters is neglected in the existing parametric formulae. As seen in Equation (A4), only the parameter *r/t* was taken into consideration in the research of Lawrence et al. [29]. The influence of other parameters, instead, was considered by a constant coefficient *β*.

To investigate the accuracy of the SCFs predicted by the parametric formulae listed in Table 1, a total of 1600 cases (i.e., 400 cases for each type of joints and loading conditions) were created based on the spline model. The parameters for the spline model were generated randomly and independently within the ranges specified in Section 2.4. The SCF was then calculated by means of Equations (A1)–(A4) in Table 1 and compared with the results carried out by the FE analyses. The ratio of deviation *δ*_2_ (%), that specifies the errors between the FE analyses and parametric formulae is also defined in Equation (3), where *K_t,formulae_* and *K_t,FEM_* denoted the SCF calculated by the formulae and in the FE analyses, respectively.
(3)$$\delta_{2}=\left(K_{t,formulae}-K_{t,FEM}\right)/K_{t,FEM}$$

As can be seen in Figure 5, in many cases, the parametric formulae proposed by Brennan et al. [17], Molski [24], and Lawrence et al. [29] resulted in inaccurate predictions. The SCFs assessed by those authors are smaller than the ones obtained in FE analyses based on the spline model as a whole. This is consistent with the explanation about the differences between the line model and spline model given in Section 2.3. The SCFs proposed by Molski [24] illustrated a comparatively higher precision than other authors which could be explained by the high order of formulae as seen in Equation (A3), while certain deviations could still be observed in some cases. Although the parametric formula provided by Monahan [28] was also derived from the line model, the SCF calculated by Monahan’s formula was, in many cases, higher than the results of FE analyses. This is because the parameter T0/t in Monahan’s weld model was kept constant with a value of 2.8, which is higher than half of the corresponding parameter randomly generated in this research. According to analyses in Section 3.2, the SCF increased with the increase of *T/t* and *L/t*, the sum of which is *T*_0_/*t*. Therefore, the SCF calculated by Monahan’s is higher than the SCF obtained in FE analyses when other parameters are identical. The above-mentioned influence of the geometric parameters on the variability of SCF is analyzed in the subsequent section.

Moreover, the deviation analyses in Figure 5a,b pointed out the small ranges of the input parameters adopted in the paper by Brennan et al. [17] and Monahan [28]. The parameters within the application ranges indicated a comparatively smaller deviation than those outside the application ranges. Even so, a particular portion of these points exceeded the limit error (±5% deviation) defined in this paper. In Figure 5c, the decline of deviation could be observed with increasing of parameter *ρ/a* and *T/a*, which can serve as a reference for the application range of this parametric formula. As can be seen in Figure 5d, the deviation decreased with the increase of *r/t*, implying that the formula proposed by Lawrence et al. [29] could provide accurate results with large weld toe radius and thin plate thickness. In conclusion, the existing parametric formulae showed insufficient precisions for the input parameter both within and outside the ranges of application. Therefore, the present work aims to propose a new formula considering the streamline shape of fillet weld.

### 3.2. Influence of Parameters on the SCF

Before dealing with the fitting of the parametric formulae, it is necessary to investigate the variability of the SCF with the geometric parameters. Moreover, some parameters defined in the spline model might have little influence on the SCF, and therefore, they could be neglected in the subsequent fitting of parametric formulae. Ten cases were selected for each parameter where the investigated parameter is varied linearly within specified ranges while the other parameters were kept constant. It is worth emphasizing that the SCF in this paper was calculated by the notch stress to obtain conservative results. The trends of SCF for T-shape joint and cruciform joint under tensile and bending stresses are shown in Figure 6a–i.

It could be observed that the parameters related to the shape of bottom weld toe (*r/t*, *θ*) have the most significant influence on the SCFs in all the four configurations of Section 2.2 (i.e., T-shape and cruciform joints under tensile or bending loading conditions). The SCFs of cruciform joints under the tensile stress were always greater than the SCFs of other welded joints with the same parameters. This aspect indicates that the cruciform configuration represents the worst-case scenario in terms of fatigue performance. The regression analysis on the training data is more likely to converge with fewer input variables. Therefore, if the change of the SCF is within 5% for the whole range of values assumed by a geometrical parameter, the dependency of the SCF on that geometric parameter can be neglected in the subsequent fitting formulae to improve the convergence of the regression analysis. It is assumed that a 5% error on the estimation of the SCF would still be acceptable and within the considered deviation. Basic mathematics functions, with the greatest R-squared, were selected to have an analytic expression of the SCF as a function of the geometry. The fitting curves for each parameter are reported in Figure 6. The fitting curves of the parameters that have little influence on the SCF under certain conditions were not drawn and kept constant in the subsequent regression analyses. According to the results of the regression analysis, the greatest R-squared could be achieved when the power function was used for the parameter *r*_1_/*t*, and the cubic function was used for the other parameters. The basic form of the chosen parametric formulae is reported in Equation (4)
(4)$$K_{t}=1+cons\cdot f\left(T/t\right)\cdot f\left(r_{1}/t\right)\cdot f\left(\theta_{1}\right)\cdot f\left(L_{1}/t\right)\cdot f\left(L_{2}/t\right)\cdot f\left(H/t\right)$$
where *cons* indicates a constant that has to be defined to achieve a better convergence.

### 3.3. Parametric Formulae of SCF

The regression analysis was carried out considering 400 cases, as training data, for each condition mentioned in Section 3.1. It is believed that the total number of simulations would represent a significant set to define a reliable parametric formula based on the spline model. The notch stress at the weld toe was obtained from the FE model, and SCF was fitted by the basic functions shown in Equation (4). Additional 200 cases, with parameters randomly generated, were built as testing data by FE models to examine the accuracy of the proposed formulae. The results of the training data system and testing data system are shown in Figure 7. The R-squared was also calculated for T-shape joint and cruciform joint under tensile stress and bending stress.

As seen in Figure 7, considering the ranges of variation of the geometrical parameters discussed before, most of the results are concentrated between a value of 1.5 and 4. The data with SCFs higher than 4 are minimal since high SCFs usually occurred in correspondence of a very small range of *r/t* values close to zero (see Figure 6b). This can explain why, even with a random selection of the parameters within the application ranges, a low probability of high SCFs occurrence is expected. The deviation between the results obtained by the proposed parametric formulae and by the FE analyses was smaller than 5%, indicating that this novel approach can lead to more accurate results than the existing formulae. The coefficients for each parametric formula are listed in Appendix B.

## 4. Probability Analysis of Fillet Weld Based on the Parametric Formulae

Section 3.3 dealt with the proposition of a parametric formula for the evaluation of the SCF for a fillet weld in T-shape joint and cruciform joint based on the spline model. The severity of the stress concentration of these two welded joints is compared in this section. Based on the data from the training system and testing system with broad application ranges, the mean values of the SCFs for the four investigated configurations were 2.134, 2.103, 2.420, and 1.908, respectively. The stress concentration, assessed by the mean values, is highest in the cruciform joint under the tensile stress. However, the application of mean values to assess the severity of stress concentration could not be representative. This is due to the fact that the mean values can be affected by the presence of high SCFs generated by geometrical parameters that lie within small ranges. For instance, as discussed in the previous section, the high SCFs obtained for small values of the *r/t* ratio can influence the mean value significantly.

As mentioned earlier, the fatigue performance of fillet weld is strongly related to the SCF. As well, the SCF is a function of the geometric parameters that can be affected during the welding process. Even in the same welding seam, the shape of the fillet weld could be quite different. It is important to evaluate the SCF of a certain fillet weld to be able to apply some treatments and improve the shape in case the SCF is particularly high. Employing the probability analysis, it is possible to obtain a probabilistic distribution of the SCFs under certain welding conditions. On the other hand, since the parameters in the training and testing systems were randomly generated following a uniform distribution, they are not suitable for probability analyses. Actual measures should be used instead.

Here, the geometric data of the cruciform joint specimen mentioned in Section 2.3 were measured for the probability analysis of fillet weld. The welding conditions were listed in Table 2. It is assumed the same welding conditions were applied to a T-shape joint. A total of 50 sections were cut by the 3D Scanner VL-300. Thus, 200 cases for the fillet weld could be obtained, and parameters based on the spline model were measured.

The measured parameters used in the parametric formulae to obtain the SCFs of the T-shape joint and cruciform joint under the tension and bending stresses. The SCFs of 200 cases for each configuration are displayed in Figure 8. It is believed the cases are sufficient to achieve the distribution rules of SCF for different fillet welds. The SCF is reported in the abscissa, whereas in the ordinate it is reported the corresponding number of cases distributed within a certain range. It could be easily observed that the histograms of the SCFs resemble the normal distribution, even if a perfect normal distribution could not be achieved due to the limited number of data. The lognormal density function, applicable to statistical data greater than 0 (see Equation (5)), was applied to assess the probability distribution since the SCF is always greater than unity.
(5)$$f(K_{t},\mu ,\sigma )=\left\{\begin{array}{c} \frac{1}{K_{t}\sigma \sqrt{2\pi }}\exp(-\frac{\ln{\left(K_{t}-\mu \right)}^{2}}{2\sigma^{2}}),K_{t}>0\\ 0\;\;\;\;,K_{t}\leq 0 \end{array}\right\}$$
where, *K_t_* denoted the SCF in the probability analysis, *μ* denoted the mean value of SCF, and *σ* denoted the standard deviation.

The probability analysis was conducted on the four geometrical configurations of Section 2.2. The characteristic values of *μ* and *σ,* together with the probability distribution curve were also added in Figure 8. Considering the mean value *μ*, it could be concluded that worst scenario case, in terms of stress concentration, is represented by the cruciform joint under the tensile stress, followed by the T-shape joint under the bending stress, the cruciform joint under the bending stress, and lastly by the T-shape joint under the tensile stress. Moreover, the calculated mean values of the SCFs using the measured geometrical parameters were quite different from those assessed by the weld parameters generated randomly. This result indicated that it is crucial to assess the stress concentration based on the real welding conditions.

The possible ranges where the SCF might be distributed were also essential to judge the possible SCFs for the fillet welds. Considering a survival probability of 95%, the confidence intervals for the four configurations were also calculated. In detail, the values of probable SCFs are within the range of [2.032, 2.348] for T-shape joint under the tension stress, [2.350, 2.678] under the bending stress, [2.589, 2.913] for cruciform joint under tensile stress, and [2.051, 2.376] under the bending stress. The range of overlap of the confidence intervals for the T-shape joint under the tensile stress and cruciform joint under the bending stress is quite wide, indicating a similar response in terms of stress concentration. The SCF of the cruciform joint under tensile stress is the highest even under the probability approach. This conclusion is valid for the welding condition used in this research; moreover, the same method could be used based on the parametric formulae proposed in this study for assessment of the SCF under other welding conditions.

## 5. Conclusions

The parametric studies on the notch stress concentration factor at the weld toe of fillet weld were carried by a new weld model. The parametric formulae for SCF of T-shape joints and cruciform joints under tensile and bending stress were proposed based on a large set of FE analysis. A wide application range of parameters was considered as follows:(1)Stiffener thickness *T/t*: 0.3–2.0;(2)Bottom weld toe radius *r*_1_/*t*: 0.003–0.36;(3)Top weld toe radius *r*_2_/*t*: 0.003–0.36;(4)Bottom flank angle *θ*_1_: 20°–90°(5)Top flank angle *θ*_2_: 20°–90°(6)Bottom weld leg length *L*_1_/*t*: 0.3–2.0;(7)Top weld leg length *L*_2_/*t*: 0.3–2.0;(8)Salient point position *1/n*: 0.2–0.9; and(9)Hump height *H/t*: 0.0–0.3.

The spline model was proposed in this research to achieve higher precision for the assessment of SCF since it considers a better approximation of the real fillet weld shape. The line model was proved to give an inaccurate prediction of the SCF at weld toe of the fillet weld if the streamline shape of real fillet weld was considered. All the parametric formulae were proposed based on the regression analyses of numerous training data. The deviation ratio of the SCF evaluated by the proposed parametric formula and compared with the FE analyses results were proved to be less than 5%, according to the testing data system. Considering the welding condition in this research, the stress concentration of the cruciform joint under tensile stress conditions represents the worst case scenario if assessed by the confidence interval of 95% survival probability. Further research will be carried out on the fillet weld with initial angular distortion, which is inevitable in the welded structures. Moreover, the 3D spline model is going to be considered as the SCF on different positions of the weld might have interactive influence on each other.

## Figures and Tables

**Figure 1 materials-13-04639-f001:**
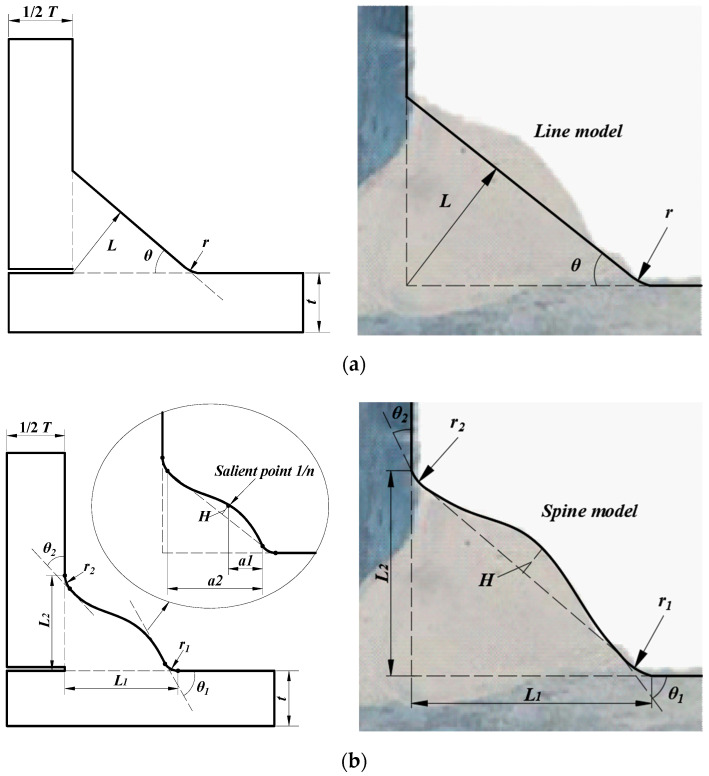
Fillet weld model (T-shape joint): (**a**) line model; (**b**) spline model.

**Figure 2 materials-13-04639-f002:**
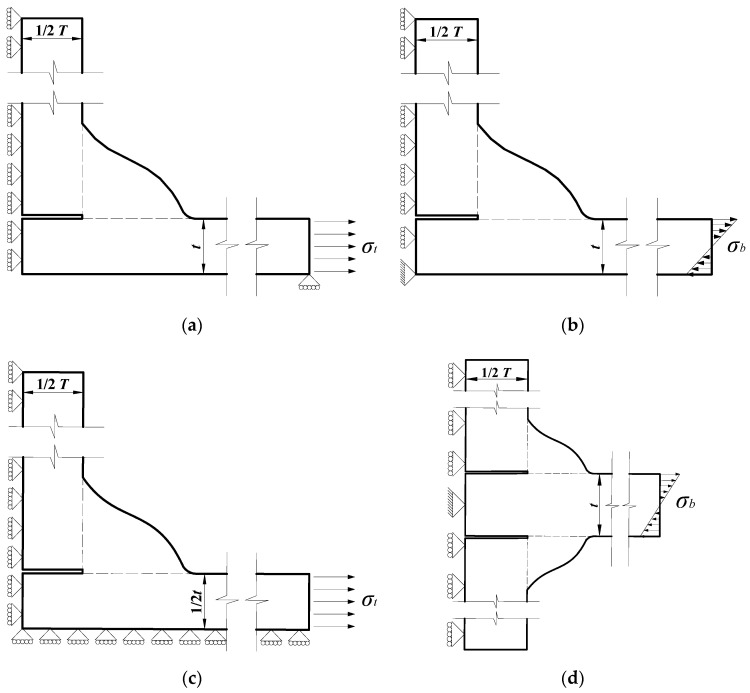
Boundary and loading conditions: (**a**) T-shape joint (tensile stress); (**b**) T-shape joint (bending stress); (**c**) cruciform joint (tensile stress); (**d**) cruciform joint (bending stress).

**Figure 3 materials-13-04639-f003:**
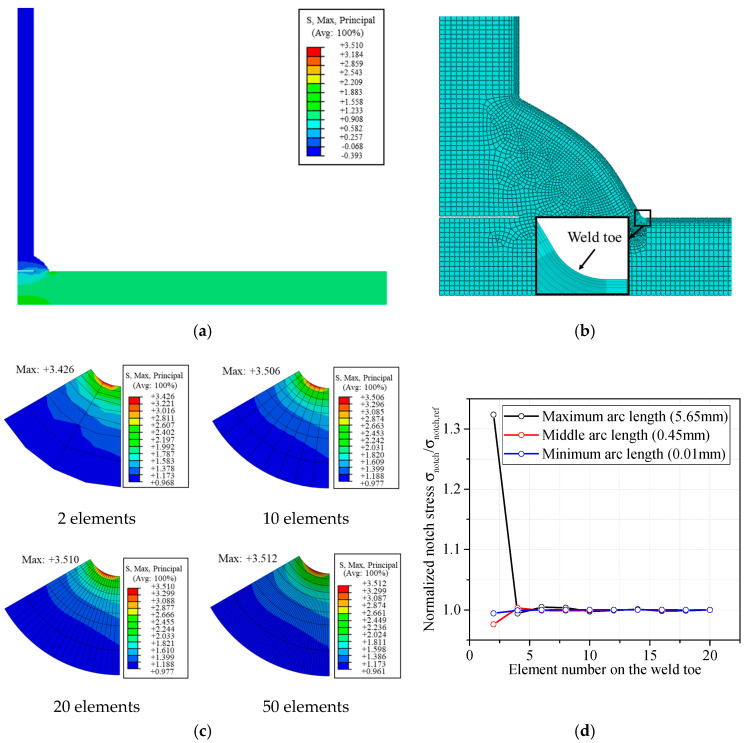
Finite element model (unit: MPa): (**a**) global model (**b**) typical global and local meshes; (**c**) mesh discretization; (**d**) convergence analysis.

**Figure 4 materials-13-04639-f004:**
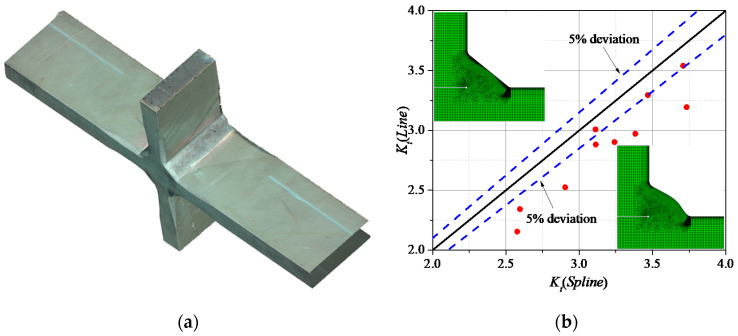
Comparison between the spline model and line model: (**a**) scanned cruciform specimen; (**b**) deviation ratio.

**Figure 5 materials-13-04639-f005:**
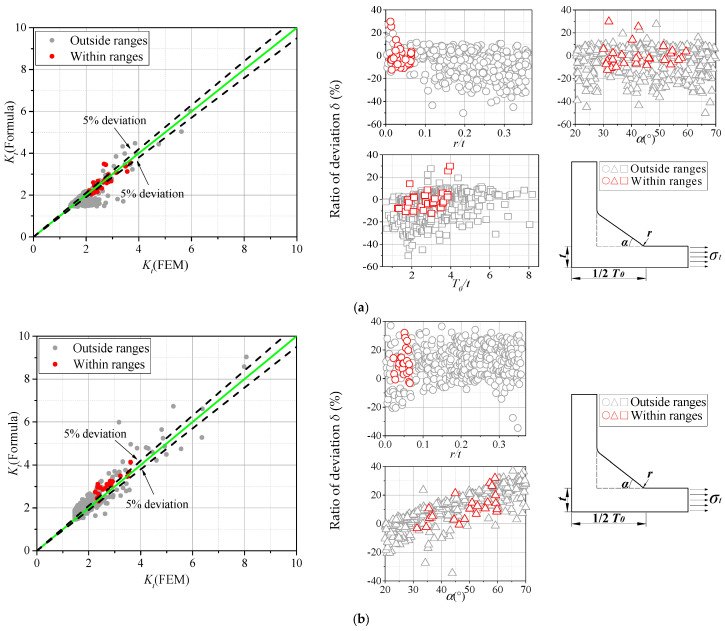
Accuracy of existing formulae: (**a**) Brennan et al. [17]; (**b**) Monahan [28]; (**c**) Molski et al. [24]; (**d**) Lawrence et al. [29].

**Figure 6 materials-13-04639-f006:**
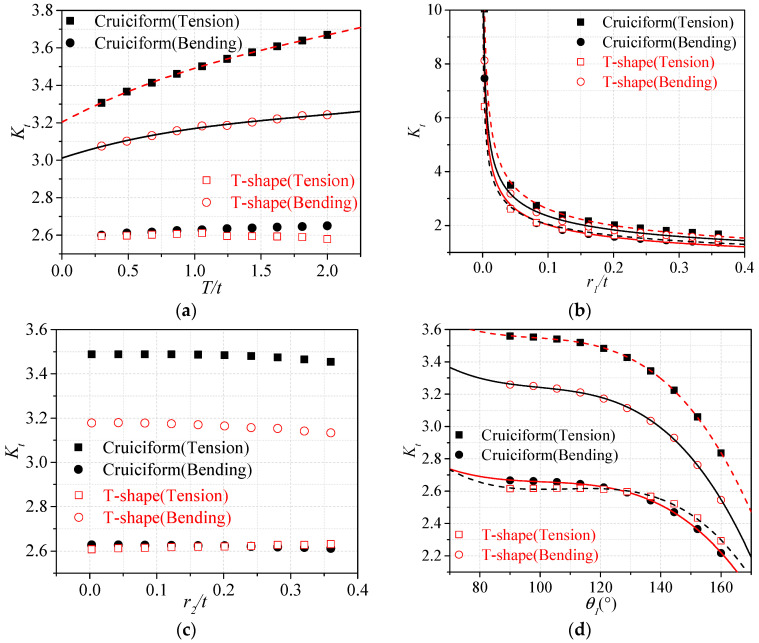
Accuracy of existing formulae: (**a**) stiffener thickness, *T/t*; (**b**) bottom weld toe radius, *r*_1_/*t*; (**c**) top weld toe radius, *r*_2_/*t*; (**d**) bottom flank angle, *θ*_1_; (**e**) top flank angle, *θ*_2_; (**f**) bottom weld leg length, *L*_1_/*t*; (**g**) top weld leg length, *L*_2_/*t*; (**h**) salient point position, *1/n*; (**i**) hump height, *H/t*.

**Figure 7 materials-13-04639-f007:**
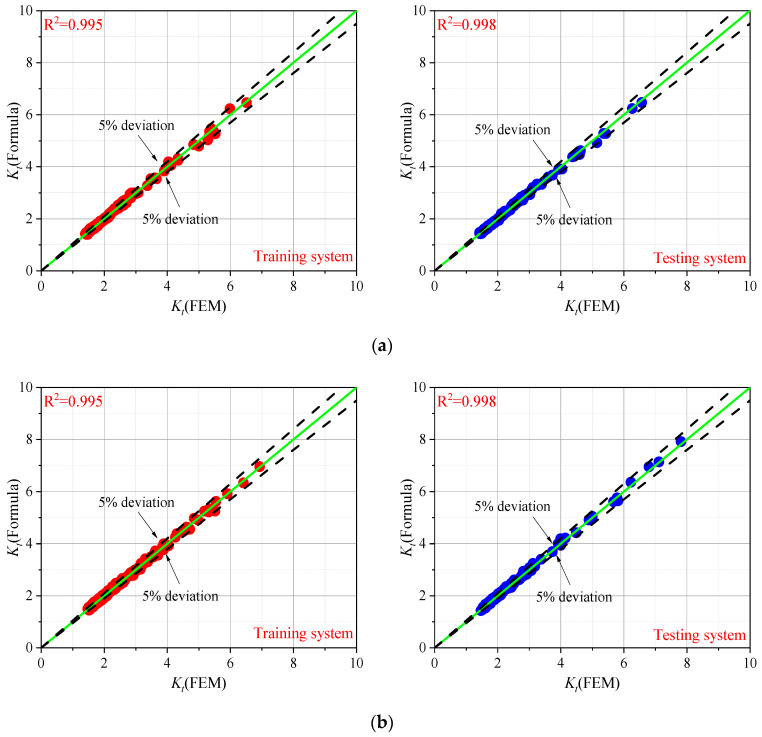
Comparison of SCF determined by FE analysis and proposed parametric formulae: (**a**) T-shape joint under tensile stress (**b**) T-shape joint under bending stress; (**c**) cruciform joint under tensile stress; (**d**) cruciform joint under bending stress.

**Figure 8 materials-13-04639-f008:**
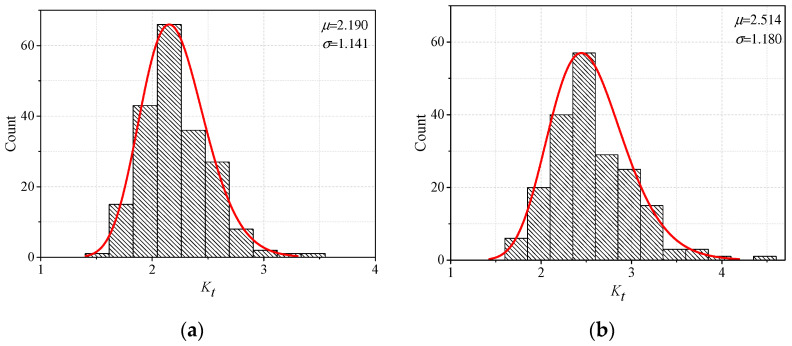
Probability distribution of SCF: (**a**) T-shape joint under tensile stress (**b**) T-shape joint under bending stress; (**c**) cruciform joint under tensile stress; (**d**) cruciform joint under bending stress.

**Table 1 materials-13-04639-t001:** Typical parametric formulae for SCF of the T-shape joints and cruciform joints.

References	Parametric Formulae	Application Ranges	Remarks
Brennan et al. [17]	Equation (A1)	*r/t*: 0.01–0.066 *α*: 30°–60° *T*_0_/*t*: 0.3–4.0	T-shape joint under tensile stress
Monahan [28]	Equation (A2)	*r/t*: 0.02–0.066 *α*: 30°–60°	T-shape joint under bending stress
Molski [24]	Equation (A3)	*ρ/a*: 0.0–1.3 *a/t*: 0.0–1.3 *T/a*: 1.0–4.0	Cruciform joint under tensile stress
Lawrence et al. [29]	Equation (A4)	*r/t*: 0.03–0.25	Cruciform joint under bending stress

**Table 2 materials-13-04639-t002:** Welding conditions.

Welding Method	Welding Position	Welding Wire	Welding Current	Welding Voltage	Welding Speed	Interpass Temperature
GMA(CO_2_)	Downhand	MX-Z200	270 A	34 V	26.1 cm/min	Below 80 °C

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
