# Peer review of "Parametric Formula for Stress Concentration Factor of Fillet Weld Joints with Spline Bead Profile"

_materials, 2020, doi:10.3390/ma13204639_

Round 1

Reviewer 1 Report

Dear Authors,

Thank You for choosing Materials for Your publication. I have read your paper with interest. The problem of weld fatigue strength is nowadays key in many branches of industry. That’s why the improving the quality of fatigue life prediction I consider as one of the most important subjects. You work has very good structure and should be clear for readers. My only concern is about testing the quality of FE mesh. You tested mesh discretization in tangential direction while the greatest stress gradient can be observed in radial direction. I would rather change the size of elements in radial direction with reasonable size of elements in tangential direction. Nevertheless the stress distribution is smooth and mesh for 10 and 20 elements is quite regular (no spaghetti shape) so discretizion is not the issue.

I appreciate the fact that in contrast to many previous works You really presented analyses of mesh quality. I recommend the paper for publication in present form however I have found  4 points that could be improved:

In Fig 5. graphs' ordinates in right column  are not defined,

In line 332 there should be Fig.7 instead of Fig.8,

In line 382 You have mistaken the Molski formula (You have copied the Monahan formula instead of Ktt=X^−0.3264(A0t+A1tX+A2tX^2+A3tX^3+A4tX^4)κt(T/a,X,Y)),

and finally in line 460 You missed author’s name for reference 28. It should be Monahan.

Best regards

Author Response

Dear reviewer:

Thank you very much for your valuable comments. It’s really a great help to improve the quality of this paper. We’ve revised the paper according to your comments and the questions are answered. As there are figures included in the response, please turn to the attachment for the specific information. 

Best regards

Reviewer 2 Report

Overall the study is good prepared and can be interesting for Materials journal readers. There is nothing major that I felt needed to comment beside two minor issues:

a/ Line 192: typographical error (T0/t  change to T0/t),

b/ Figure 5: please consider to add the y-axis label to all graphs.

Author Response

(The authors gave the same response as above.)

Reviewer 3 Report

In my opinion, the article provides interesting solutions concerning fillet weld joints with spline bead profile. Research carried out by using numerical and experimental way. The research was based mainly on the prepared numerical model. The work considers several different cases of solutions. The figures and description of the work look good, but a few small points could be taken into account:

  1. The paper should present definitely more results of the FEM analysis, although in the form of stress maps within the tested systems, because only few results were presented - and the work focuses mainly on numerical research. Please add a few interesting figures presenting the results of the tests in addition to the few presented in the paper.
  2. Please present the global result within the framework of the numerical model, not only in the context of the selected areas, but the distribution of the stresses in the entire modelled construction.
  3. The conclusions should be expanded slightly as there is no reference to further research plans.

Article can be published after changes some minor problems.

Author Response

(The authors gave the same response as above.)
